# Foliated fracton order from gauging subsystem symmetries

**Wilbur Shirley[1⋆], Kevin Slagle[1,2] and Xie Chen[1]**

**1** Department of Physics and Institute for Quantum Information and Matter,
California Institute of Technology, Pasadena, California 91125, USA
**2** Department of Physics, University of Toronto, Toronto, Ontario M5S 1A7, Canada

⋆ wshirley@caltech.edu

## Abstract

Based on several previous examples, we summarize explicitly the general procedure to gauge models with subsystem symmetries, which are symmetries with generators that have support within a sub-manifold of the system. The gauging process can be applied to any local quantum model on a lattice that is invariant under the subsystem symmetry. We focus primarily on simple 3D paramagnetic states with planar symmetries. For these systems, the gauged theory may exhibit foliated fracton order and we find that the species of symmetry charges in the paramagnet directly determine the resulting foliated fracton order. Moreover, we find that gauging linear subsystem symmetries in 2D or 3D models results in a self-duality similar to gauging global symmetries in 1D.



# 1   Introduction

Gauging is a powerful tool in the study of gapped quantum phases with global symmetry. When gauging the global symmetry of a system, gauge fields corresponding to the symmetry group are added to the system so that the global symmetry can be enhanced to a local symmetry. It is useful to consider such a procedure because different phases under global symmetry map into different phases of the gauge theory. Symmetric (e.g. paramagnetic) phases map into deconfined gauge theories while symmetry breaking phases map into a Higgsed gauge theory. Different symmetry protected topological (SPT)/symmetry enriched topological (SET) phases map into different deconfined gauge theories with different statistics among the gauge fluxes (see, e.g., Refs. [1, 2]).

Recently, it has been realized that a similar gauging procedure can be applied to systems with subsystem symmetries as well [3–9]. Subsystem symmetries are symmetries with generators that act non-trivially only on a sub-manifold of the system. After gauging, the system is mapped to a model with 'fracton order' [3, 10–37]. This relation has been demonstrated for various classical/quantum spin models, stabilizer codes, domain-frame condensate models, etc. In this paper, we summarize and make explicit the general gauging procedure. That is, we describe explicitly a systematic procedure for gauging models with subsystem symmetries which can be applied to any local quantum model with such symmetry. In particular, the gauge fields are added at the center of 'minimal' coupling terms which are not on-site symmetric and which generate all other non-on-site-symmetric coupling terms. A modified Hamiltonian can then be written with enhanced local symmetry and with dynamical terms for the gauge field, which defines the gauge theory. We focus on abelian symmetry groups only in this paper.

The next key question is: what is the relation between the ungauged order under subsystem symmetry and the gauged fracton order? To address this question, we study the mapping between ungauged and gauged phases (several of these examples have been studied in the previous literature [3–9]) and propose a way to interpret the correspondence. In 2D and 3D, gauging linear subsystem symmetries (which act on 1D lines) maps paramagnetic (trivially symmetric) phases and symmetry breaking phases into one another, while subsystem symmetry protected topological (SPT) phases [6] may map into themselves. This is similar to the case of global symmetries in 1D, where paramagnets are mapped into symmetry breaking phases, and SPT phases can map into SPTs. In 3D, gauging planar subsystem symmetries leads to foliated fracton order, as defined in Refs. [15, 28]. In particular, symmetry charges that transform under planar symmetries in one, two or three directions map directly to planon, lineon and fracton charge excitations, which are restricted to move only in a plane, along a line, or which

cannot move at all. The restricted motion of the charge excitations in the fracton model hence originates from the requirement to preserve subsystem symmetries in the ungauged model. By counting the species of symmetry charges in the ungauged model, we can make direct connection to the foliated fracton order after gauging. For example, it was shown in Ref. [3] that gauging the (paramagnet phase of) the plaquette Ising model and the tetrahedral Ising model results in the X-cube and the checkerboard model respectively. By counting symmetry charges, we can see that the checkerboard model should be equivalent to two copies of the X-cube model. We present the mapping between the two in Ref. [38] and in section 4, we explain how counting symmetry charges leads to the same conclusion.

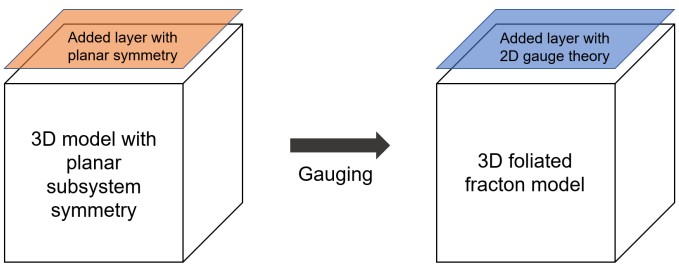

Figure 1: Correspondence of foliation structure in 3D systems with planar subsystem symmetry and 3D foliated fracton models.

Given the analogous foliation (or layered) structure in 3D models with planar subsystem symmetry and 3D foliated fracton phases, there is a natural correspondence. As shown in Fig. 1, for 3D models with planar subsystem symmetry, to increase the system size by one lattice spacing in the direction of one set of planar subsystem symmetries, it is necessary to add degrees of freedom (DOFs) on an entire plane and increase the number of generators of subsystem symmetries by one. The added planar subsystem symmetry acts as a global symmetry on the added plane. On the other hand, as we discussed in Ref. [15, 28], for 3D foliated fracton phases, to increase the system size by one lattice spacing along one of the foliation axes, it is necessary to add a layer containing a gapped 2D topological state as a resource. Thus, it is natural that subsystem symmetry symmetric states gauge into foliated fracton models since the added layer gauges into a deconfined 2D gauge theory with gapped topological order.

The paper is organized as follows: In section 2, we briefly review the procedure of gauging global symmetries using as an example the 2D paramagnetic state. Section 3 then discusses the generalized gauging procedure that can be applied to systems with subsystem symmetries in a systematic way. Multiple examples (including examples that have appeared in the previous literature) are discussed to show how the procedure works in different situations. Section 4 studies the correspondence between phases with subsystem symmetries and the phases of their gauged theories through multiple examples and the result is summarized in Table 1 in section 5.

## 2 Review: Gauging global symmetry

First, we give a brief review of the procedure for gauging global symmetries (for more careful discussions see, e.g., Refs. [1, 39]). We consider the simplest example: the transverse field Ising model with global $Z_2$ symmetry, coupled to a $Z_2$ gauge field. The Hamiltonian takes the simple form of

$$H = -J_x \sum_v \sigma_v^x - J_z \sum_{\langle vw \rangle} \sigma_v^z \sigma_w^z, \tag{1}$$

where the $\sigma$'s are Pauli matrices on each lattice site (blue dots in Fig. 2) and $\langle vw \rangle$ denotes nearest neighbor pairs. The system has a global $Z_2$ symmetry of $U = \prod_v \sigma_v^x$. To couple the model to a $Z_2$ gauge field, we introduce gauge field degrees of freedom $\tau$ on each link of the lattice (green dots in Fig.2). $\tau^x$ corresponds to (the exponential $e^{iE}$ of) the 'electric field' of the gauge field and $\tau^z$ corresponds to (the exponential of) the 'vector potential' of the gauge field. The local symmetry, or the Gauss's law, is given by $A_v = \sigma_v^x \prod_{e \ni v} \tau_e^x$ where the product is over all edges $e$ with $v$ as one end point.

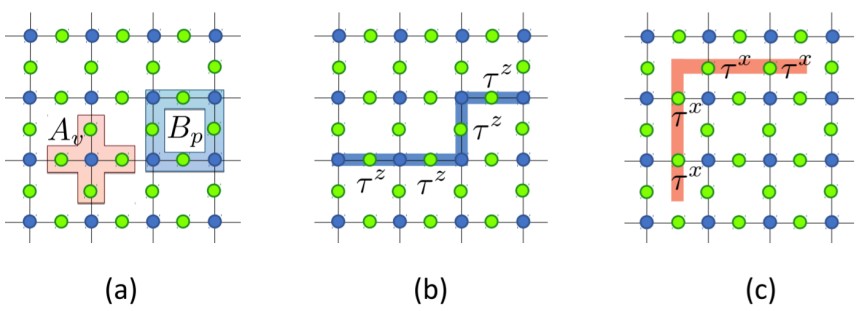

Figure 2: Gauging global $Z_2$ symmetry in 2D. **(a)** Vertex $A_v = \sigma_v^x \prod_{e \ni v} \tau_e^x$ and plaquette $B_p = \prod_{e \in p} \tau_e^z$ terms that appear in Eq. (2). **(b-c)** String operators.

Next, we couple $H$ to the gauge fields such that the new Hamiltonian is invariant under the local symmetry transformations $A_v$. The transverse field terms $\sigma_i^x$ are already invariant under the local symmetries, so we do not need to modify them and simply include them in the new Hamiltonian. The Ising coupling terms $\sigma_i^z \sigma_j^z$ need to be replaced with $\sigma_i^z \tau_{ij}^z \sigma_j^z$ in order to be gauge invariant (i.e. commute with the $A_v$ term). Besides that we add the vertex term $A_v = \sigma_v^x \prod_{e \ni v} \tau_e^x$ at every vertex $v$ to enforce gauge symmetry (Gauss's law) and $B_p = \prod_{e \in p} \tau_e^z$, where the product is over all edges around a plaquette $p$, to enforce the zero flux constraint on every plaquette. The total Hamiltonian then reads

$$H_g = -J_x \sum_v \sigma_v^x - J_z \sum_{\langle vw \rangle} \sigma_v^z \tau_{vw}^z \sigma_w^z - J_v \sum_v \sigma_v^x \prod_{e \ni v} \tau_e^x - J_p \sum_p \prod_{e \in p} \tau_e^z. \tag{2}$$

When $J_z = 0$, the Ising model $H$ is in the symmetric paramagnetic phase. After gauging, it maps to the deconfined phase of the $Z_2$ gauge theory. This can be seen by noticing that when the energy of the $\sum_v \sigma_v^x$ term is minimized, the gauged Hamiltonian reduces to

$$H_{\mathrm{TC}} = -J_v \sum_v \prod_{e \ni v} \tau_e^x - J_p \sum_p \prod_{e \in p} \tau_e^z, \tag{3}$$

which is exactly the toric code Hamiltonian representing the deconfined phase of the $Z_2$ gauge theory. The low energy excitations include a bosonic gauge flux, which corresponds to the violation of one $\prod_{e \in p} \tau_e^z$ term, and a bosonic gauge charge, which corresponds to the violation of one $\prod_{e \ni v} \tau_e^x$ term. These two excitations can be created with string operators shown in Fig. 2b-c. They braid with each other with a phase factor of $-1$, which is the Aharonov-Bohm phase factor in the $Z_2$ case.

When $J_x = 0$, the Ising model $H$ is in the symmetry breaking ferromagnetic phase. After gauging, it maps to the Higgsed phase which lacks non-trivial topological order. This can be seen by noticing that when $J_x = 0$, $H_g$ has a unique ground state and no fractional excitations.

This gauging procedure can be applied to any local quantum Hamiltonian on any lattice satisfying a global symmetry $G$ by introducing gauge fields on the links of the lattice, enforcing gauge symmetry (Gauss's law), modifying interaction terms to be gauge invariant, and finally

including a flux term for the gauge field. By doing so, we obtain a gauge theory of group $G$. The properties of the gauge theory can be determined from the ungauged model in the following ways:

1. If the symmetry is spontaneously broken in the ungauged model, then the gauge theory is Higgsed with trivial topological order.

2. Otherwise, the deconfined gauge charge comes from the symmetry charge. The deconfined charges are either bosonic or fermionic, depending on whether the symmetry charges in the ungauged model are bosonic or fermionic.

3. The deconfined gauge flux comes from the symmetry flux, except it is dynamical. The statistics of the gauge flux depends on the particular order (SPT/SET) of the ungauged model. Some interesting examples include: gauging the $Z_2$ fermion parity symmetry in the 2D chiral $p + ip$ superconductor results in a non-abelian flux; also, gauging the 2D bosonic SPT with $Z_2$ symmetry results in a semionic flux.

4. The braiding statistics between a gauge charge and a gauge flux is independent of the original order; it is given by the Aharonov-Bohm phase factor, which is determined by the symmetry group. For example, in a $Z_N$ gauge theory, the phase factor between an elementary charge and an elementary flux is $e^{i2\pi/N}$.

5. In 1D, gauge theories are not topologically ordered. Symmetry breaking and trivial SPT phases map into each other upon gauging. Non-trivial SPT phases can map to themselves upon gauging. (We briefly review the gauging of 1D phases in appendix B.)

## 3   Gauging subsystem symmetry: general procedure

How do we gauge models with subsystem symmetries? The simplest example of a system with subsystem symmetry is an Ising paramagnet on a cubic lattice (corresponding to the plaquette Ising model in Ref. [3]). Consider a cubic lattice with spin 1/2 degrees of freedom at each lattice site (blue dots in Fig. 3). The Hamiltonian is simply given by $H = \sum_v \sigma_v^x$. This Hamiltonian is invariant under planar subsystem symmetries

$$U_n^{XY} = \prod_{v \in P_n^{XY}} \sigma_v^x, \qquad U_n^{YZ} = \prod_{v \in P_m^{YZ}} \sigma_v^x, \qquad U_n^{ZX} = \prod_{v \in P_n^{ZX}} \sigma_v^x, \qquad (4)$$

where $P_n^{XY}$ labels the $XY$ plane with $Z$ direction coordinate $n$ and similarly for $P_n^{YZ}$ and $P_n^{ZX}$. Throughout this paper, we use $X$, $Y$, $Z$ to label spatial directions and $x$, $y$, $z$ to label spin directions.

This model (with additional plaquette terms) was originally considered in Ref. [3]; however, we are not including the Ising coupling term here for simplicity of discussion. To gauge it, Ref. [3] proposed to add a gauge degree of freedom $\tau$ at each face-center of the cubic lattice (green dots in Fig. 3). The gauge symmetry is then given by

$$A_v = \sigma_v^x \prod_{f \ni v} \tau_f^x, \qquad (5)$$

which is the product of a symmetry charge $\sigma_v^x$ at a site $v$ and the (twelve) electric gauge fields $\tau_f^x$ on the neighboring faces $f$. The gauge flux terms, which are minimal pure vector potential terms that satisfy the gauge symmetry, now involve the product of four $\tau^z$'s as shown in Fig. 3. The gauged Hamiltonian takes the form

$$H_g = -\sum_v \sigma_v^x - \sum_v A_v - \sum_c \left( B_c^{XY} + B_c^{YZ} + B_c^{ZX} \right). \qquad (6)$$

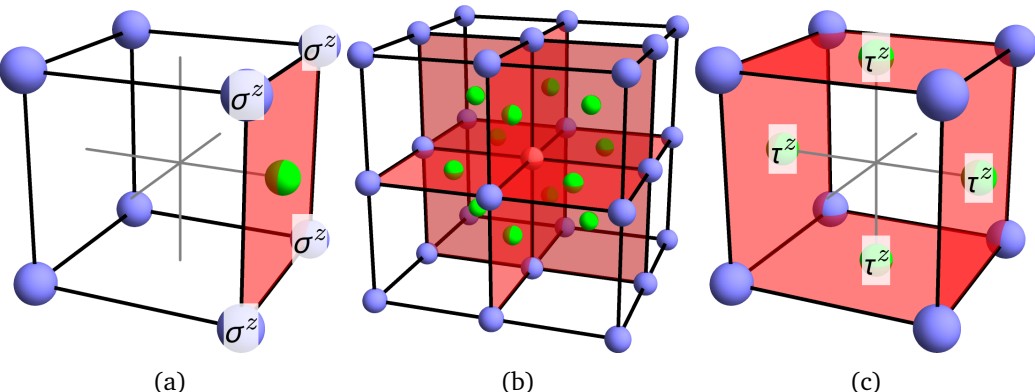

Figure 3: Gauging planar symmetry on a cubic lattice. **(a)** The minimal symmetric coupling term: a product of four $\sigma^z$ around a plaquette (of the cubic lattice of blue spheres). A gauge field $\tau$ (green sphere) is therefore placed at the center of the plaquette, and all other plaquettes. The gray lines form the dual lattice. **(b)** The red vertex is involved in the twelve minimal coupling terms highlighted by red squares. The gauge symmetry term is a product of a $\sigma^x$ at the red sphere and twelve $\tau^x$ on the green spheres. **(c)** The product of four minimal coupling terms around the four blue plaquettes is the identity. The corresponding flux term is a product of four $\tau^z$ on the green spheres.

Since the symmetry charges are fixed by the transverse field $\sigma^x$ (in the ground state), the zero temperature phase of the gauged Hamiltonian becomes equivalent to that of the X-cube model [3].

However, for generic systems with subsystem symmetry the degrees of freedom may be located at different places in the lattice and may transform under the subsystem symmetry in different ways. For example, in Ref. [3], an example was discussed where the ungauged model contains DOFs at the vertices *and* at the face centers of a cubic lattice, where the subsystem symmetry acts on planes with integer and half integer coordinates (in units of the cubic lattice constant). Ref. [8] discussed an example where the DOFs lives both at the vertices and body centers; the ones at vertices transform under subsystem symmetry in one direction only. For a generic configuration of lattice structure and DOFs, where should the gauge fields be added and how should the gauge symmetry of the gauged model be defined?

## 3.1 General procedure

We will now outline a gauging procedure that is consistent with the gauging procedure for global symmetry [1, 39] and various previous works for gauging subsystem symmetries. The input to the procedure is a lattice of degrees of freedom (in a Hilbert space), a set of symmetry operators, and a model $H = \sum h$ that is symmetric under the symmetry. We will focus on abelian groups only in this paper.

Suppose that the on-site symmetry charge at each site is measured by $\sigma_\nu^x$ (in general the charge does not have to be a $Z_2$ charge, although we use the $\sigma$ notation without loss of generality). The procedure is as follows:

1. Find the minimum coupling terms $c$ that a) are not on-site symmetric; b) are a tensor product of operators carrying elementary symmetry charges at each site; and which, c) together with on-site symmetric terms, can be composed into any coupling term satisfying the symmetry. (Note that these minimum coupling terms are not necessarily included

in the Hamiltonian; they are used only to locate the gauge degree of freedom in the next step.)

2. Assign a gauge degree of freedom $\tau_c$ at the center of each minimum coupling term. ($\tau_c^x$ can be thought of as the exponential $e^{iE}$ of the electric field $E$, while $\tau_c^z$ is the exponential of the vector potential. $\tau$ can be a general gauge field, not just a $Z_2$ one.)

3. The gauge symmetry is given by $A_v = \sigma_v^x \prod_{c \ni v} \tau_c^x$, where the product is over all minimum coupling terms $c$ that contain $v$.

4. All symmetric coupling terms $h$ can then be made into gauge symmetric terms $h_g$ by multiplying each minimal coupling factor in $h$ by a $\tau_c^z$.

5. The minimum coupling terms will usually not be independent of each other. Or sometimes, gauge fields are added for non-minimum coupling terms as well. In such cases, we then find independent minimum sets $\mathcal{C}$ of coupling terms $c \in \mathcal{C}$ whose product is either the identify or a product of on-site symmetric terms $\sigma^x$.[1] Correspondingly, the product $B_{\mathcal{C}} = \prod_{c \in \mathcal{C}} \tau_c^z$ becomes the flux term of the gauge field if it is a local term.

In this way, we can gauge a model $H = \sum h$ with global or subsystem symmetry into a gauge theory $H_g = \sum h' - \sum_v A_v - \sum_{\mathcal{C}} B_{\mathcal{C}}$. Note that a large part this procedure, such as determining the gauge degrees of freedom and the gauge symmetry, is completely independent of the original Hamiltonian and depends on the action of the symmetry operators on the ungauged Hilbert space. The only step that depends on the original Hamiltonian is step 4 where the original Hamiltonian is made gauge symmetric.

Let us consider some examples to see how this works.

## 3.2 Example: global symmetry

For global symmetry, the minimum symmetric coupling term is a nearest neighbor two-body term of the form $O_i O'_j$ where $O_i$ carries charge $e$ and $O'_j$ carries charge $-e$. Other symmetric coupling terms, including non-nearest-neighbor two-body terms and multi-body terms, can all be constructed as composites of the nearest-neighbor two-body terms and on-site symmetric terms. Therefore, the gauge DOFs are assigned to each link of the lattice. The gauge symmetry term involves one lattice site and all the emanating links. The set of two-body terms around the same plaquette combine into on-site symmetric terms; therefore we have one flux term per plaquette. This is exactly the gauging procedure we reviewed in Sec. 2. Changing the lattice structure corresponds to choosing a different set of minimum coupling terms, which does not affect the nature of the gauge theory obtained.

## 3.3 Example: 3D planar symmetry on a cubic lattice

For the subsystem symmetry example discussed above (DOFs at vertices of cubic lattice, transforming under planar symmetry in three directions), the minimum symmetric coupling term is the four-body plaquette term $\prod_{v \in p} \sigma_v^z$, as shown in Fig. 3a. All other symmetric coupling terms can be obtained as composites of such plaquette terms and on-site symmetric terms. Therefore, as suggested in Ref. [3], we can add one gauge field per plaquette. Each vertex is involved in 12 minimum coupling terms; therefore the gauge symmetry term is a product of one $\sigma^x$ and twelve $\tau^x$ (Fig. 3b). Four minimum coupling terms around the same cube combine into identity as shown in Fig. 3c; therefore we have the corresponding flux terms. This is exactly the gauging procedure we reviewed at the beginning of this section [Sec. 3].

---

[1]Products of on-site symmetric terms can result for example when choices of minimal couplings terms contain $\sigma^x$.

### 3.4 Example: 3D planar symmetry on a FCC lattice

Consider the situation corresponding to the tetrahedral Ising model discussed in Ref. [3], as shown in Fig. 4. Besides the DOF $\sigma_v$ at vertices of the cubic lattice, there are DOF $\sigma_f$ at the faces of the cubic lattice. Subsystem symmetry acts on each $XY$, $YZ$ and $ZX$ direction plane either with integer or half integer coordinates.:

$$U_n^{XY} = \prod_{v,f \in P_n^{XY}} \sigma_v^x \sigma_f^x, \qquad U_n^{YZ} = \prod_{v,f \in P_n^{YZ}} \sigma_v^x \sigma_f^x, \qquad U_n^{ZX} = \prod_{v,f \in P_n^{ZX}} \sigma_v^x \sigma_f^x,$$
$$U_{n+1/2}^{XY} = \prod_{f \in P_{n+1/2}^{XY}} \sigma_f^x, \qquad U_{n+1/2}^{YZ} = \prod_{f \in P_{n+1/2}^{YZ}} \sigma_f^x, \qquad U_{n+1/2}^{ZX} = \prod_{f \in P_n^{ZX}} \sigma_f^x. \tag{7}$$

The minimum coupling terms, as shown in Fig. 4a, are the tetrahedral terms involving one $\sigma_v^z$ and three $\sigma_f^z$'s. All other symmetric coupling terms, including four-body terms of $\sigma_v^z$'s and four-body terms of $\sigma_f^z$, can all be constructed from this minimum coupling term. Therefore, as discuss in Ref. [3], one gauge DOF $\tau$ is to be assigned to each tetrahedron. The gauge symmetry terms are the product of one $\sigma^x$ together with the eight $\tau^x$'s in the surrounding tetrahedrons (Fig. 4b). The product of the same eight tetrahedron minimum coupling terms also happens to be the identity; therefore, we have the product of eight $\tau^z$'s as the flux term (Fig. 4b). If the $\sigma$ DOF are all polarized by $-\sigma^x$, the gauged model becomes exactly the same as the checkerboard model.

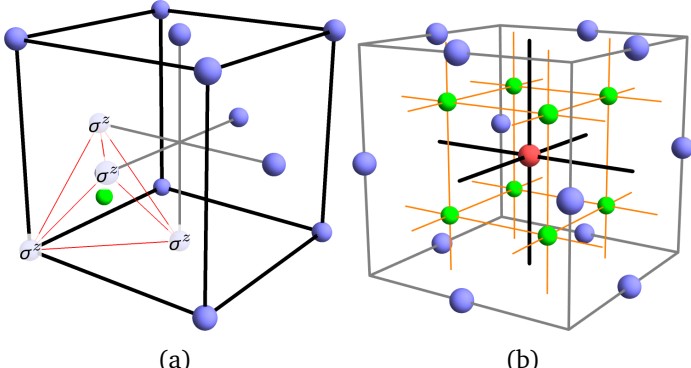

(a)            (b)

Figure 4: Gauging planar symmetry on FCC lattice. **(a)** A minimal symmetric coupling term: a product of four $\sigma^z$ at the corners of the red tetrahedron. A gauge field $\tau$ (green sphere) is placed at the center of tetrahedron. Within the above cube, there are eight tetrahedra (one for each corner of the cube) and gauge fields. The gray lines form the dual lattice. **(b)** The red vertex is involved in the eight minimal coupling tetrahedron terms centered at the green spheres. The gauge symmetry term is thus a product of a $\sigma^x$ at the red sphere and eight $\tau^x$ on the green spheres. The product of the eight minimal coupling tetrahedron terms is the identity. The corresponding flux term is a product of eight $\tau^z$ on the green spheres.

### 3.5 Example: 3D planar symmetry on a BCC lattice

Now consider the situation described in Ref. [8], where there is one DOF $\sigma_0$ at each cube center and three DOFs $\sigma_a$, $\sigma_b$, $\sigma_c$ at each vertex, as shown in Fig. 5. $\sigma_0$ transforms under subsystem planar symmetries in all three directions while $\sigma_a$, $\sigma_b$, and $\sigma_c$ transform only under symmetries in $YZ$, $ZX$, and $XY$ planes, respectively. An $XY$-plane subsystem symmetry generator is a product of all $\sigma_0^x$ in a particular $XY$ plane ($P_{m+1/2}^{XY}$) with $Z$ coordinate $m + 1/2$

and all $\sigma_{\mathrm{c}}^{x}$ in the two neighboring $XY$ planes ($P_m^{XY}$ and $P_{m+1}^{XY}$) with $Z$ coordinate $m$ and $m+1$:

$$U_{m+1/2}^{XY} = \prod_{i \in P_{m+1/2}^{XY}} \sigma_{0,i}^x \prod_{j \in P_m^{XY}} \sigma_{\mathrm{c},j}^x \prod_{k \in P_{m+1}^{XY}} \sigma_{\mathrm{c},k}^x. \tag{8}$$

$U^{YZ}$ and $U^{ZX}$ are defined in similar ways. The minimum coupling terms are the triangular terms shown in Fig. 5a. All other symmetric coupling terms can be composed from these minimum coupling terms. Therefore, to gauge the model, we need to assign one gauge DOF $\tau$ per triangle. The gauge symmetry terms are then the product of one $\sigma_0^x$ with 24 $\tau^x$'s around it (Fig. 5b), and the product of one $\sigma_a^x$ (or $\sigma_b^x$, $\sigma_c^x$) with four $\tau^x$'s around it (Fig. 5c). The product of four triangular coupling terms is the identity, therefore we have the product of the corresponding four $\tau^z$'s as the flux term (Fig. 5d). This is the minimum gauging scheme for such a distribution of symmetry charges.

We could add gauge fields corresponding to non-minimum coupling terms as well. This is what was done in Ref. [8], where a gauge field is added for each four-body plaquette coupling term of the $\sigma_0$'s. Since this four-body term can be obtained by composing two triangular terms, this results in one more type of gauge flux term corresponding to the product of the $\tau^z$ associated with these three coupling terms (one plaquette and two triangular terms).

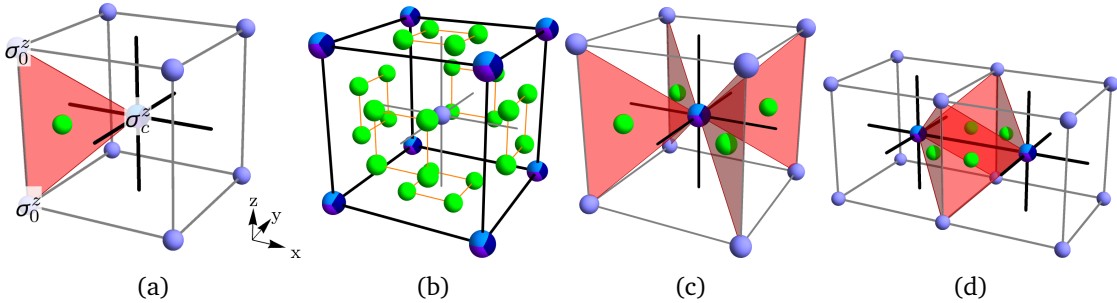

$$\text{(a)} \qquad\qquad \text{(b)} \qquad\qquad \text{(c)} \qquad\qquad \text{(d)}$$

Figure 5: Gauging planar symmetry on BCC lattice. **(a)** A minimal symmetric coupling term: a product of two $\sigma_0^z$ and one $\sigma_c^z$. The black lines form the cubic lattice, while the gray lines form the dual lattice. There are 12 minimal coupling terms within the shown dual-lattice cube: one for each gray edge of the cube. For the other terms, the $\sigma_c^z$ at the center becomes a $\sigma_a^z$ or $\sigma_b^z$ when the two $\sigma_0^z$ are displaced in the $x$ or $y$ direction, respectively. **(b)** The body-center is involved in $4 \times 6$ minimal coupling terms, which are centered at the green spheres, which lie on the faces of the black cube. (The orange lines are guides for the eye.) The gauge symmetry term is therefore a product of a $\sigma^x$ at the center and 24 $\tau^x$ on the green spheres. **(c)** The $\sigma_c^z$ operator in the center is involved in 4 minimal coupling terms (highlighted in red). The gauge symmetry term is therefore a product of a $\sigma_c^x$ at the center and four $\tau^x$ on the green spheres. **(d)** The product of the four minimal coupling triangular terms is the identity. The corresponding flux term is a product of four $\tau^z$ on the four green spheres.

## 4 Correspondence before and after gauging

Using the general gauging procedure, in this section we are going to explore the correspondence between models with subsystem symmetry (before gauging) and the gauged model with (potential) foliated fracton order. We refer to such a correspondence as the 'gauging correspondence'. While the following discussion is mostly based on specific examples, we expect

several features of the gauging correspondence to apply generically, as specified below. In Appendix A, we will also show that the gauging procedure can be applied to global dipole conservation symmetries to produce a symmetric tensor gauge theory.

## 4.1 Planar symmetry and foliated fracton order

First, let's discuss models with subsystem planar symmetries. We are going to study models of increasing complexity – paramagnets with subsystem planar symmetries in one direction, two directions, three directions and four directions respectively – as well as models where the symmetries are spontaneously broken. We expect the following features to be generically true in the gauging correspondence: 1. when the planar symmetries are spontaneously broken, the gauged model does not have nontrivial order; 2. when the planar symmetries are not spontaneously broken, the gauged model has foliated fracton order 3. symmetry charges transforming under planar symmetries in one direction, two directions, and three or more directions turn into planon excitations, lineon excitations, and fracton excitations respectively upon gauging. The first feature is analogous to the Higgs mechanism in usual gauge theories. For the second one, we gave an intuitive understanding in the introduction section. Let us briefly discuss the third one before moving onto examples.

In Ref. [53], we proposed to characterize fractional excitations in foliated fracton phases using the notion of *quotient superselection sectors* (QSS). In particular, two fractional excitations are considered as equivalent (i.e. they belong to the same QSS class) if they differ only by local excitation and planons – a fractional excitation that moves in a 2D plane. Among the foliated fracton phases that we have studied, there are two types of QSS:

1. fracton sectors where the fractional excitation is fully immobile as an individual quasiparticle, and

2. lineon sectors where the excitation can only move along a straight line.

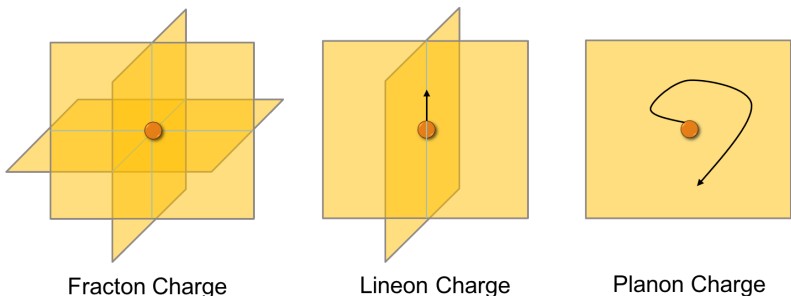

Fracton Charge     Lineon Charge     Planon Charge

Figure 6: Symmetry charges transforming under planar symmetries in three, two, one directions are fractons (cannot move), lineons (can move only along a line), and planons (can move only in a plane) respectively.

In terms of the gauging correspondence, it is easy to see how the fracton/lineon QSS can emerge after gauging subsystem symmetries. Before gauging, if a symmetry charge transforms under planar subsystem symmetries in three directions, then to preserve subsystem symmetry, this charge cannot move freely in any direction. It is pinned at the intersection point of the three planes, as shown in Fig. 6, and such fracton symmetry charges have to be created four at a time. Upon gauging, they become the fracton gauge charges. If a symmetry charge transforms under planar symmetries in two directions, then this charge can move but only along the intersection line of the two planes. Such lineon symmetry charges become the lineon gauge charge upon gauging. Finally, if a symmetry charge transforms under planar symmetries

in one direction only, then this charge can move along the plane. Such planon symmetry charges become the planon gauge charge upon gauging. Composites of fracton charges can become lineon or planon charges. For example, composing two $Z_2$ fracton charges in the same plane and displaced by a diagonal direction results in a lineon charge because the composite carries nontrivial symmetry charge in the two orthogonal planes only. By analyzing how the symmetry charges and their composites transform under subsystem symmetry, we can see how the gauging correspondence emerges. Let us see how this works through the following examples.

### 4.1.1 3D paramagnet with planar symmetry in one direction

We start with a simple and almost trivial case where the subsystem symmetry acts only in $XY$ planes. Consider again the cubic lattice with DOF at vertices and the paramagnetic model $H = -\sum_v \sigma_v^x$. The subsystem symmetry is given by

$$U_m^{XY} = \prod_{v \in P_m^{XY}} \sigma_v^x. \tag{9}$$

Upon gauging, this model should naturally map to a stack of 2D (untwisted) deconfined gauge theories in the $XY$ plane. The symmetry charges become the planon gauge charges in each 2D layer. The gauged theory is a trivial foliated fracton phase. Of course, this result does not depend sensitively on the lattice structure or details of the Hamiltonian, as long as the planar symmetries are preserved.

### 4.1.2 3D paramagnet with planar symmetry in two directions

A less trivial example is the 3D paramagnet $H = -\sum_v \sigma_v^x$ with two sets of planar symmetries

$$U_m^{XZ} = \prod_{v \in P_m^{XZ}} \sigma_v^x, \ \ U_n^{YZ} = \prod_{v \in P_n^{YZ}} \sigma_v^x. \tag{10}$$

Each symmetry charge transforms under planar symmetries in two directions and hence becomes a lineon gauge charge upon gauging. The combination of two symmetry charges separated in the $X$ or $Y$ directions transform under planar symmetry in one direction only and hence is a planon. The combination of two symmetry charges separated in the $Z$ direction does not transform under subsystem symmetry at all and hence is a not a fractional excitation. Therefore, in the gauged theory, we expect only one lineon QSS in the charge sector.

This can be seen explicitly by applying the gauging procedure described in section 3. The two minimum coupling terms are 1) four $\sigma^z$'s around a plaquette in the same $XY$ plane (Fig. 7a), and 2) two $\sigma^z$'s along the $Z$ axis (Fig. 7b). Correspondingly, gauge fields are placed in each $XY$ plane plaquette and on each link in the $Z$ direction. The gauge symmetry term involves the product of one $\sigma_v^x$, four $\tau_{XY}^x$'s and two $\tau_Z^x$'s, as shown in Fig. 7c. The product of two plaquette coupling terms and four link coupling terms is identity, giving rise to the flux term as shown in Fig. 7d. The gauge charge, which corresponds to the violation of the gauge symmetry term, is a lineon that moves in the $Z$ direction. It turns out that the flux excitation is also a lineon that moves in the $Z$ direction. This is the anisotropic model introduced in Ref. [53].

### 4.1.3 3D paramagnet with planar symmetry in three directions

Now let us consider the case where the planar subsystem symmetries lie along three directions. We have discussed the gauging procedure of three different cases (with different distributions

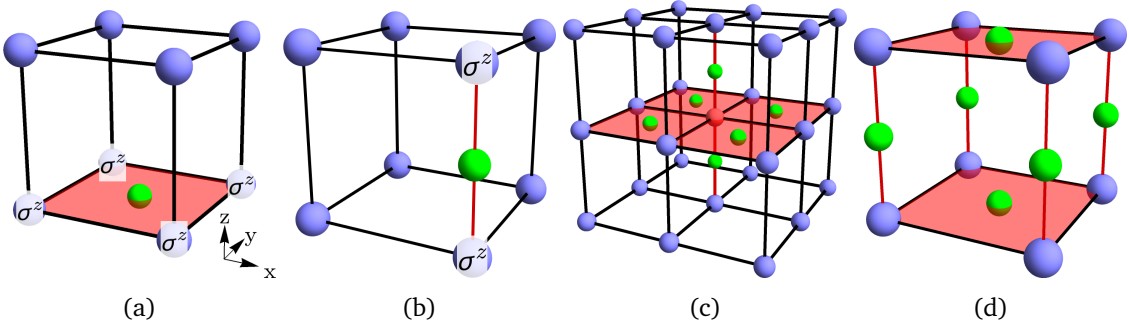

Figure 7: Gauging planar symmetry in $XZ$ and $YZ$ directions only. **(a-b)** Minimal coupling terms. **(c)** The red vertex term in the center is included in four plaquette minimal coupling terms (red plaquettes) and two Z-axis terms (red edges). Therefore, the gauge symmetry term is a product of a $\sigma^x$ at the center (red sphere) and six $\tau^x$ at the green spheres. **(d)** The flux term is a product of six $\tau^z$ at the green spheres.

of symmetry charges) in section 3. Now we will examine how the symmetry charge becomes a gauge charge through the gauging process and how the corresponding foliated fracton order emerges after gauging.

## A. Cubic lattice

In the case discussed in section 3.3, where symmetry charges live at the vertices of a 3D cubic lattice and transform under planar symmetries in all three directions, each symmetry charge is a fracton and cannot move (since the charge is conserved on every plane). If two symmetry charges separated in the $X$, $Y$ or $Z$ direction are combined, then the composite transforms under planar symmetry in one direction only and hence is a planon. Therefore, upon gauging, the gauge charge sector of the gauge theory should contain only one quotient superselection sector – a fracton QSS. This is indeed the case for the corresponding gauge theory of X-cube model. As discussed in Ref. [53], the X-cube model contains three elementary QSSs: one fracton QSS and two lineon QSS. The one fracton QSS is the gauge charge sector of the gauge theory while the two lineon QSSs are the gauge flux sector of the gauge theory.

## B. Cubic lattice: dual model

In fact, the X-cube model can be obtained through gauging a different model. Consider a 3D cubic lattice with two DOFs $\sigma_r$ and $\sigma_b$ (red and blue) at each lattice site. The red $\sigma_r$ transform under planar symmetry in $XY$ and $YZ$ directions; the blue $\sigma_b$ transform under planar symmetry in $YZ$ and $ZX$ directions; and their composite at each lattice site transforms under planar symmetry in $ZX$ and $XY$ directions. That is, the symmetries act as

$$U_m^{XY} = \prod_{v \in P_m^{XY}} \sigma_{v,r}^x, \qquad U_m^{YZ} = \prod_{v \in P_m^{YZ}} \sigma_{v,r}^x \sigma_{v,b}^x, \qquad U_n^{ZX} = \prod_{v \in P_n^{ZX}} \sigma_{v,b}^x. \tag{11}$$

The minimum coupling terms are two-body terms $\sigma_{v,r}^z \sigma_{v+\hat{y},r}^z$ in the $Y$ direction, two-body terms $\sigma_{v,b}^z \sigma_{v+\hat{z},b}^z$ in the $Z$ direction, and four-body terms $\sigma_{v,r}^z \sigma_{v,b}^z \sigma_{v+\hat{x},b}^z \sigma_{v+\hat{x},r}^z$ in the $X$ direction, as shown in Fig. 8a. Therefore, according to the general procedure, a gauge field is added to each link of the cubic lattice. The gauge symmetry term is the product of $\sigma_{v,r}^x$ ($\sigma_{v,b}^x$) with four $\tau^x$ on neighboring links in the $XY$ plane ($ZX$ plane), as shown in Fig. 8b-c. The combination of twelve minimum coupling terms around a cube is identity, therefore the flux term is the product of twelve $\tau^z$ around a cube as shown in Fig. 8d.

If the $\sigma$ spins are all polarized by Hamiltonian $H = -\sum_v \left( \sigma_{v,r}^x + \sigma_{v,b}^x \right)$, then the gauged model is exactly the X-cube model, but as the electromagnetic dual of the previous case. The symmetry charges transform under two planar symmetries, and therefore gauge into two independent lineon gauge charges (that move in the $Y$ and $Z$ directions). Their combination is a lineon charge that transforms under the $XY$ and $XZ$ planar symmetries and therefore moves only in the $X$ direction. If two red charges separated in the $X$, $Y$, or $Z$ directions are combined, then they form either a planon or a local excitation, and similarly for the blue charges. Therefore, the gauge charge sector contains two independent lineon QSSs. The gauge flux in this case makes up the fracton QSS.

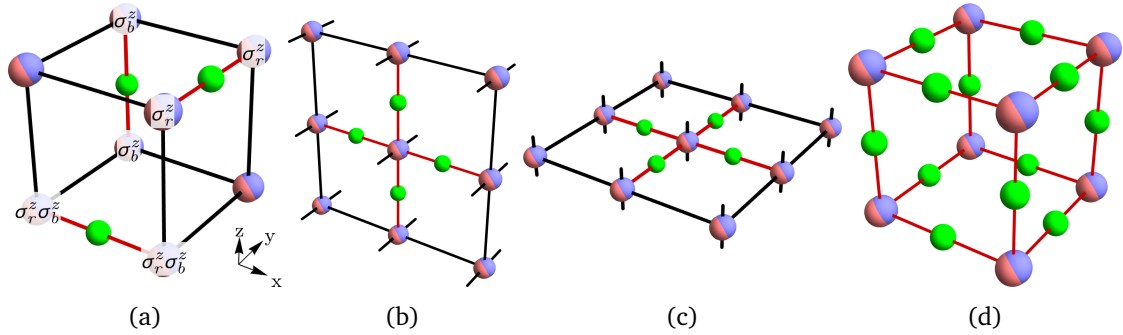

Figure 8: Gauging planar symmetry on the cubic lattice with lineon charges. **(a)** The three minimal coupling terms, which are each a product of $\sigma^z$ operators across one of the red links. **(b)** A $\sigma_b^z$ operator at the center is included in four minimal coupling terms on the red links. The corresponding gauge symmetry term is a product of a $\sigma_b^x$ at the center and $\tau^x$ operators on the green spheres. **(c)** Same as (b), but for $\sigma_r^z$. **(d)** The flux term is a product of $\tau^z$ on the twelve green spheres.

### C. FCC lattice

In the second case discussed in section 3.4, symmetry charges live both at vertices and face centers and transform under planar symmetry in all three directions. Again, each symmetry charge (both the vertex and face-center charges) is a fracton and cannot move. The combination of two vertex charges separated in $X$, $Y$, or $Z$ directions transforms under planar symmetry in one direction only, and hence is a planon. Therefore, the vertex charge alone makes one fracton QSS after gauging. The combination of a vertex charge and a face-center charge separated by half of a face diagonal transforms under two planar symmetries and are hence lineons. Similarly, the combination of two face-center charges separated by half of a face diagonal (out of the plane of the face) are also lineons. Taking into account neutral excitations – excitations carrying no symmetry charges – involving one vertex charge and three face-center charges, we can see that there are all together two independent lineon sectors. Therefore, upon gauging, the charge sector should contain one independent fracton QSS and two independent lineon QSSs. This corresponds exactly to the combination of the original and dual cubic lattice examples discussed above. Therefore, the gauged theory – the checkerboard model [3] – should be equivalent to two copies of X-cube model combined in a electromagnetic dual way. This is exactly what we show in Ref. [38].

### D. BCC lattice

Now we come to the case discussed in section 3.5, where symmetry charges at cube center transform in three directions while symmetry charges at vertices transform in one direction only. The vertex charges are planon charges so they can be omitted when counting QSSs.

The cube center charge is a fracton. Two fracton charges separated in the $X$, $Y$ or $Z$ direction combine into a planon. Therefore, upon gauging, the gauge charge sector contains only one fracton QSS. If the ungauged Hamiltonian is in the trivial phase (given for example by $H = -\sum_i \sigma^x_{0,i} - \sum_j \sigma^x_{a,j} - \sum_k \sigma^x_{b,k} - \sum_l \sigma^x_{c,l}$ ), then the gauged model would belong to the same foliated fracton phase as the X-cube model.

In Ref. [8], a twisted version of the ungauged Hamiltonian is discussed. Upon gauging, the charge sector remains the same, while the flux sector may have different statistics compared to the X-cube model. Ref. [8] discussed the difference in statistics in terms of the self rotation of lineons. In Ref. [53], we show that this difference can be removed if 2D layers of twisted gauge theories are added to the 3D fracton model. Therefore, the gauged model has the same foliated fracton order as the X-cube model. Correspondingly, the difference between the twisted and non-twisted versions of the ungauged Hamiltonian can be removed by adding 2D layers of twisted SPTs. Therefore, the twisted ungauged model is equivalent to a 'weak SSPT', i.e. a stack of 2D SPTs, as defined in Ref. [6].

### 4.1.4 3D paramagnet with planar symmetry in 4 directions

It is also possible to construct a paramagnet in which every DOF transforms under a planar subsystem symmetry in 4 different directions. The model is constructed as follows: first, a lattice is constructed out of a fourfold foliation structure. To be precise, given four stacks of parallel planes such that no four planes intersect at a single point, a natural cellulation structure is defined in which each elementary 3-cell is a polyhedron bounded by these planes. Then, a $\sigma$ DOF is placed in each 3-cell. The planar subsystem symmetries act on all 3-cells *between* neighboring parallel planes. The minimal symmetric coupling terms are the four-body terms $\prod_{v \in p} \sigma^z_v$ with a $\sigma^z$ operator on each of the four 3-cells adjacent to a given edge (which is along the intersection between two planes). In the dual cellulation (or lattice), this edge is dual to a quadrilateral plaquette $p$, and the 3-cells are dual to vertices $v$. Upon gauging, the subsystem symmetric paramagnet defined on this type of lattice yields a generalized X-cube model as discussed in Ref. [16]. For example, using this type of construction, one can obtain the stacked kagome lattice X-cube model.

### 4.1.5 3D symmetry breaking state with planar symmetry

In all previous examples, for the ungauged model, we considered the simplest symmetric Hamiltonian of the form $H = -\sum_v \sigma^x_v$ where the ground state is symmetric under all subsystem symmetries. For global symmetry, it is known that when the matter field undergoes spontaneous symmetry breaking, the gauge field is Higgsed and the gauge theory become non-topological. For subsystem symmetry, a similar Higgs mechanism applies, as first discussed in Ref. [3]. Let us repeat the exercise and see how Higgsing occurs in the cubic lattice example of section 3.3.

The minimum Ising coupling term that can be added to the system is the plaquette term involving four $\sigma^z$'s (Fig. 3a). To make the term gauge invariant, we attach a $\tau^z$ term in the middle of the plaquette. The total gauged Hamiltonian hence takes the form

$$H_g = -\sum_p \tau^z_p \prod_{v \in p} \sigma^z_v - \sum_v A_v - \sum_c \left( B^{XY}_c + B^{YZ}_c + B^{ZX}_c \right). \tag{12}$$

The $B_c$ terms are actually redundant for determining ground state because they can be composed out of the plaquette terms. Therefore, the Hamiltonian can be simplified into

$$H_g = -\sum_p \tau^z_p \prod_{v \in p} \sigma^z_v - \sum_v \sigma^x_v \prod_{v \in p} \tau^x_p. \tag{13}$$

This is a cluster state [54] Hamiltonian where the $\sigma$ and $\tau$ DOFs are connected through face diagonals. It has a unique ground state, and hence no topological or fracton order.

## 4.2 Linear symmetry and duality

Now let's consider subsystem linear symmetries in 2D and 3D models. We find that the gauging correspondence works in a very similar way to that of linear symmetries in 1D. It is well known (and we review it in Appendix B) that upon gauging the linear (global) symmetry in 1D, the gauged model also has an emergent global linear symmetry at low energy which comes from the zero flux constraint around the 1D ring. The gauging procedure leads to a duality between trivial symmetric paramagnets and symmetry breaking phases and a (self)-duality among nontrivial symmetry protected topological phases. From the examples discussed in this section, we find a similar correspondence in 2D and 3D with subsystem linear symmetries:

1. the model after gauging has linear subsystem symmetries at low energy which comes from the zero flux constraint around nontrivial loops;

2. symmetry breaking phases are mapped to trivial paramagnets;

3. trivial paramagnets are mapped to symmetry breaking phases;

4. non-trivial subsystem symmetry protected topological phases are mapped to non-trivial subsystem symmetry protected topological phases.

We expect these features to apply generically to all models with linear subsystem symmetries.

### 4.2.1 2D paramagnet/symmetry breaking state with linear symmetry

It is possible for 2D systems to have linear subsystem symmetries. As we will see, gauging 2D systems with linear subsystem symmetries bears great similarity to gauging global symmetries in 1D. In particular, in both cases, trivial paramagnet and symmetry breaking phases are dual to each other through gauging. Consider a 2D square lattice with a $\sigma$ DOF at each vertex. The subsystem symmetries acts along each row $L_m^X$ and each column $L_n^Y$ of the square lattice:

$$U_m^X = \prod_{v \in L_m^X} \sigma_v^x, \quad U_n^Y = \prod_{v \in L_n^Y} \sigma_v^x. \tag{14}$$

The minimum coupling term satisfying these symmetries is a product of four $\sigma^z$ around a plaquette. Consider the ungauged Hamiltonian

$$H = -B_x \sum_v \sigma_v^x - J \sum_p \prod_{v \in p} \sigma_v^z. \tag{15}$$

To gauge this model, we place one gauge DOF $\tau_p$ on each plaquette so that the gauge symmetry is given by $A_v = \sigma_v^x \prod_{p \ni v} \tau_p^x$. No local flux term satisfies all of the gauge symmetries; the only allowed flux terms are products along an entire row or a column:

$$B_{m,m+1}^X = \prod_{p \in L_{m,m+1}^X} \tau_p^z, \qquad\qquad B_{n,n+1}^Y = \prod_{p \in L_{n,n+1}^Y} \tau_p^z. \tag{16}$$

Thus, the flux terms become subsystem symmetries of the gauged theory.

The Hamiltonian after gauging takes the form

$$H_g = -B_x \sum_v \sigma_v^x - J \sum_p \tau_p^z \prod_{v \in p} \sigma_v^z - J_v \sum_v \sigma_v^x \prod_{v \in p} \tau_p^x. \tag{17}$$

When $B_x = 0$, corresponding to the symmetry breaking phase before gauging, the gauged model is

$$H_g = -J \sum_p \tau_p^z \prod_{v \in p} \sigma_v^z - J_v \sum_v \sigma_v^x \prod_{v \in p} \tau_p^x, \tag{18}$$

which is a 2D cluster state model with unique ground state that is symmetric under the subsystem symmetries $B^X$ and $B^Y$. Moreover, this state can be mapped to a symmetric product state through a symmetric local unitary transformation, indicating that it is equivalent to a trivial paramagnet. The symmetric local unitary is given by

$$V = \prod_v \left( \prod_{v \in p} C_v X_p \right) \prod_v H_v \prod_v \left( \prod_{v \in p} C_v X_p \right), \tag{19}$$

where $C_v X_p = \frac{1}{2}(1 + \sigma_v^z) \otimes \tau_p^0 + \frac{1}{2}(1 - \sigma_v^z) \otimes \tau_p^x$ is the controlled-$X$ operation from a vertex spin to its neighboring gauge field and the Hadamard operator $H = \begin{pmatrix} 1 & 1 \\ 1 & -1 \end{pmatrix}$ maps between $\sigma^x$ and $\sigma^z$.

When $J = 0$, corresponding to the trivial paramagnet phase before gauging, the gauged model is

$$H_g = -B_x \sum_v \sigma_v^x - J_v \sum_v \sigma_v^x \prod_{v \in p} \tau_p^x, \tag{20}$$

which can be reduced to

$$H_g = -J_v \sum_v \prod_{v \in p} \tau_p^x, \tag{21}$$

if the $-B_x \sigma_v^x$ terms are all satisfied. This corresponds to the symmetry breaking phase of the gauge field under subsystem symmetries $B^X$ and $B^Y$.

### 4.2.2 2D linear symmetry protected topological model

We now discuss an example of a 2D model with linear SSPT order, which is self-dual under gauging the subsystem symmetries. The system contains a $\sigma$ DOF at each vertex of two interlocking square lattices labelled $\alpha$ and $\beta$. The linear symmetries act on all spins in a given row or column of either the $\alpha$ or $\beta$ lattice. Explicity, the symmetry generators are

$$U_m^{X,\alpha} = \prod_{v \in L_m^{X,\alpha}} \sigma_v^x, \qquad U_n^{Y,\alpha} = \prod_{v \in L_n^{Y,\alpha}} \sigma_v^x, \qquad U_p^{X,\beta} = \prod_{v \in L_p^{X,\beta}} \sigma_v^x, \qquad U_q^{Y,\beta} = \prod_{v \in L_q^{Y,\beta}} \sigma_v^x. \tag{22}$$

As discussed in Ref. [6], the 2D cluster state model is a strong SSPT, which exhibits a protected edge degeneracy that grows exponentially with the length of the boundary. The Hamiltonian (also shown in Fig. 9) is

$$H = -\sum_{a \in \alpha} \sigma_{i(a)}^z \sigma_{j(a)}^z \sigma_{k(a)}^z \sigma_{l(a)}^z \sigma_a^x - \sum_{b \in \beta} \sigma_{i(b)}^z \sigma_{j(b)}^z \sigma_{k(b)}^z \sigma_{l(b)}^z \sigma_b^x, \tag{23}$$

where $i(a)$, $j(a)$, $k(a)$, and $l(a)$ refer to the four $\beta$ lattice vertices neighboring vertex $a$, and vice versa for $i(b)$, $j(b)$, $k(b)$, and $l(b)$.

The minimal coupling terms satisfying the subsystem symmetry are the four-body terms around each elementary plaquette of either the $\alpha$ or $\beta$ lattice. Thus, to gauge the model, gauge fields $\tau_v$ are placed at every vertex $v$ of both the $\alpha$ and $\beta$ lattices (on top of each matter DOF), as shown in Fig. 9. The gauge symmetries then take the form $A_v = \sigma_v^x \tau_{i(v)}^x \tau_{j(v)}^x \tau_{k(v)}^x \tau_{l(v)}^x$. As

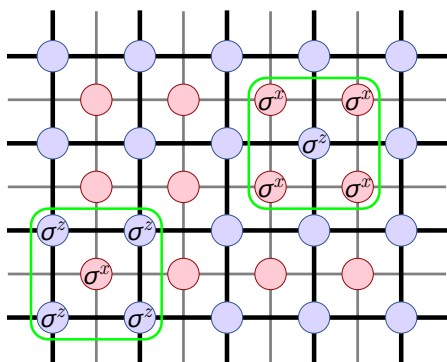

Figure 9: The 2D cluster state model. The two stabilizer terms in Eq. (23) are circled in green above. The black and gray lattices are the $\alpha$ and $\beta$ lattices. After gauging, gauge fields $\tau$ are placed on both the red and blue vertices.

in the previous example, there are no local gauge-symmetric flux operators; the only allowed flux terms act along an entire row or column:

$$B_m^{X,\alpha} = \prod_{v \in L_m^{X,\alpha}} \tau_v^z, \qquad B_n^{X,\beta} = \prod_{v \in L_n^{X,\beta}} \tau_v^z, \qquad B_p^{Y,\alpha} = \prod_{v \in L_p^{Y,\alpha}} \tau_v^z, \qquad B_q^{Y,\beta} = \prod_{v \in L_q^{Y,\beta}} \tau_v^z. \qquad (24)$$

These operators correspond to symmetry generators of the gauge theory.

Upon gauging the Hamiltonian takes the form

$$H_g = -\sum_{a \in \alpha} \tau_a^z \sigma_{i(a)}^z \sigma_{j(a)}^z \sigma_{k(a)}^z \sigma_{l(a)}^z \sigma_a^x - \sum_{b \in \beta} \tau_b^z \sigma_{i(b)}^z \sigma_{j(b)}^z \sigma_{k(b)}^z \sigma_{l(b)}^z \sigma_b^x - J_v \sum_{v \in \alpha, \beta} A_v. \qquad (25)$$

This gauged model is actually a linear SSPT and is dual to the original SSPT. To see this, note that the matter DOFs can be decoupled from the gauge DOFs via the symmetric local unitary operator

$$V = \prod_{v \in \alpha, \beta} C_{\sigma_v} X_{\tau_{i(v)}} \, C_{\sigma_v} X_{\tau_{j(v)}} \, C_{\sigma_v} X_{\tau_{k(v)}} \, C_{\sigma_v} X_{\tau_{l(v)}}, \qquad (26)$$

where as before, $C_\sigma V_\tau$ is the controlled-$X$ gate from the vertex spin $\sigma$ to an adjacent gauge field $\tau$. Then

$$V H_g V^\dagger \cong -\sum_{a \in \alpha} \tau_{i(a)}^x \tau_{j(a)}^x \tau_{k(a)}^x \tau_{l(a)}^x \tau_a^z - \sum_{b \in \beta} \tau_{i(b)}^x \tau_{j(b)}^x \tau_{k(b)}^x \tau_{l(b)}^x \tau_b^z - \sum_{v \in \alpha, \beta} \sigma_v^x, \qquad (27)$$

which is a 2D cluster state model residing on the gauge DOFs. Here the relation $H \cong H'$ indicates that $H$ and $H'$ have coinciding ground spaces and thus represent the same gapped phase.

### 4.2.3 3D models with linear subsystem symmetry

It is also possible for 3D systems to have linear subsystem symmetries. For example, suppose a system has a $\sigma$ DOF at every vertex of a cubic lattice and symmetries which act along lines of spins along the $X$, $Y$, or $Z$ direction. In this case, the minimal coupling terms that commute with the symmetries are eight-body terms $\prod_{v \in c} \sigma_v^z$ involving the 8 qubits at the corners of a cube $c$. Therefore, to gauge such models, gauge fields are placed at the centers of each cube.

The correspondence before and after gauging of linear subsystem symmetries in 3D bears similarities to the case of linear symmetries in 2D and global symmetries in 1D. For instance,

the cubic Ising Hamiltonian,

$$H = -\sum_v \sigma_v^x - \lambda \sum_c \prod_{v \in c} \sigma_v^z, \tag{28}$$

is self-dual under gauging: the weak-coupling paramagnetic phase maps into the strong-coupling subsystem symmetry breaking phase and vice versa. Furthermore, the linear SSPT given by the the 3D cluster state Hamiltonian [6] is self-dual under gauging, in analogy with the 2D cluster state linear SSPT and the 1D cluster state global SPT.

## 5 Discussion

The gauging correspondence revealed in the previous examples is summarized in the table below. Fracton charges are acted upon by planar symmetry in three directions, whereas lineon charges are acted upon by planar symmetry in two directions. The fracton and lineon charges in the table are counted up to the attachment of planon charges, which are acted upon by planar symmetry in one direction only.

Table 1: Correspondence between phases with subsystem symmetries and gauge theory phases. The X-cube and anisotropic model listed refer to the corresponding foliated fracton phase, not to the specific model.

|  | Before Gauging | After Gauging |
|---|---|---|
| Planar symmetry in 3D | One fracton charge | X-cube with lineon flux |
|  | Lineon charges in $X$, $Y$, $Z$ directions | X-cube with fracton flux |
|  | One lineon charge in $Z$ direction | Anisotropic model with lineon flux |
|  | Symmetry breaking | Topologically / fractonically trivial state |
| Linear symmetry in 2D/3D | Trivial paramagnet | Symmetry breaking |
|  | Symmetry breaking | Trivial paramagnet |
|  | Non-trivial SSPT | Non-trivial SSPT |

Therefore, by counting the types of symmetry charges before gauging, we can determine the gauge charge and correspondingly gauge flux quotient superselection sectors in the gauge theory. A highly interesting and open question is whether there are non-trivial SPT phases with planar subsystem symmetry in 3D. The model discussed in Ref. [8] we now know to be equivalent to a weak SSPT. Hence upon gauging, it gives the same foliated fracton order as the X-cube model [53]. For a truly non-trivial SSPT, upon gauging, we expect the gauge charge and gauge flux to correspond to the same quotient superselection sectors while the gauge flux has non-trivial statistics compared to the X-cube model.

## Acknowledgements

**Funding information** W.S. and X.C. are supported by the National Science Foundation under award number DMR-1654340 and the Institute for Quantum Information and Matter at Caltech. X.C. is also supported by the Alfred P. Sloan research fellowship and the Walter Burke Institute for Theoretical Physics at Caltech. K.S. is grateful for support from the NSERC of Canada, the Center for Quantum Materials at the University of Toronto, and the Walter Burke Institute for Theoretical Physics at Caltech.

# A  3D scalar charge tensor gauge theory by gauging the $U(1)$ symmetry

Section 3 considered gauging various gapped qubit models with planar symmetries. However, the gauging procedure in Sec. 3.1 can also be used to obtain the gapless $U(1)$ tensor gauge theory models [40–50], which also have fractons, lineons, and planons. In this case, the gauging procedure is very closely related to the Higgs mechanisms discussed in Ref. [51,52]. In these $U(1)$ models, one can gauge a disordered field theory that has various kinds of global charge conservations laws. Similar to the previously discussed models, the conservation laws for the $U(1)$ models also result in mobility restrictions [40].

As an example, in this section we will consider gauging the following matter Hamiltonian

$$H = \int \pi^2 + \sum_{ab}(\partial_a\partial_b\phi)^2, \tag{29}$$

which has a global symmetry that results in a conserved dipole moment

$$P^a = \int x^a \pi, \tag{30}$$

since $[H, P^a] = 0$, where $\phi$ and $\pi$ are conjugate fields: $[\phi(x), \pi(x')] = i\,\delta^3(x-x')$. In this section, Latin letters $a, b, i, j = 1, 2, 3$ denote spatial indices. Repeated indices are implicitly summed.

We will now follow the general gauging procedure. For clarity, we will number the steps to match those in Sec. 3.1.

1. The minimal coupling operators that respect the symmetry are

   $$\partial_a\partial_b\phi. \tag{31}$$

   That is, $[\partial_a\partial_b\phi, P^c] = 0$, and all other local terms that commute with $P^a$ can be written as a polynomial in $\partial_a\partial_b\phi$ and $\pi$.

2. Since the minimal coupling operator is a symmetric tensor, we introduce a symmetric tensor gauge field $A_{ab}$, which is conjugate to an electric field $E^{ab}$: $[A_{ab}(x), E^{ij}(x')] = -\frac{i}{2}(\delta_a^i\delta_b^j + \delta_a^j\delta_b^i)\delta^3(x-x')$.

3. The gauge symmetry at $x$ is $\pi(x)$ minus an electric field in place of every minimal coupling term that contains $\phi(x)$. The resulting expression can be calculated as follows

   $$\pi(x) + i\int_{x'} [\partial_a\partial_b\phi(x), \pi(x')]E^{ab}(x') = \pi(x) - \partial_a\partial_b E^{ab}(x). \tag{32}$$

4. The minimal coupling term can be made gauge symmetric by coupling it to a gauge field: $\partial_a\partial_b\phi \rightarrow \partial_a\partial_b\phi - A_{ab}$.

5. We now need to find linear combinations of the minimal coupling terms $\partial_a\partial_b\phi$ that result in zero. Equivalently, we want to find linear combinations of derivatives of $A_{ab}$ that are invariant under the replacement $A_{ab} \rightarrow A_{ab} + \partial_a\partial_b\lambda$, which is often referred to as a gauge transformation. Thus, we want to find the smallest possible basis of gauge invariant operators, which is given by the magnetic tensor $B^i_j = \epsilon^{iab}\partial_a A_{bj}$ [40].

Therefore, gauging the matter Hamiltonian [Eq. (29)] results in the following gauged Hamiltonian

$$H = \int \pi^2 + \sum_{ab}(\partial_a \partial_b \phi - A_{ab})^2 + (\pi - \partial_a \partial_b E^{ab})^2 + \sum_{ij}(\epsilon^{iab}\partial_a A_{bj})^2 + \sum_{ab}(E^{ab})^2. \tag{33}$$

$(E^{ab})^2$ is added at the end since the above model is a gapless gauge theory. Traditionally, the $(\pi - \partial_a \partial_b E^{ab})^2$ is not explicitly written, but is instead imposed as a gauge constraint or is considered irrelevant (under RG) at long length scales.

## B  Gauging global symmetry in 1D systems

In this section, we review the process of gauging 1D symmetric, symmetry breaking and SPT phases and see how symmetric and symmetry breaking phases map into each other upon gauging while SPT phases can map into themselves.

Consider the 1D transverse field Ising model with Hamiltonian

$$H = -B_x \sum_i \sigma_i^x - J \sum_i \sigma_i^z \sigma_{i+1}^z \tag{34}$$

and global symmetry $U = \prod_i \sigma_i^x$. To gauge the model, we put gauge fields $\tau$ on every link. The gauge symmetry term is $A_i = \tau_{i-1,i}^x \sigma_i^x \tau_{i,i+1}^x$. The only flux term that satisfies all the gauge symmetries is a global term $B = \prod_i \tau_{i,i+1}^z$. Therefore, the flux term effectively becomes a $Z_2$ global symmetry of the gauged model.

Coupling $H$ to the gauge field, we obtain the gauged Hamiltonian

$$H_g = -B_x \sum_i \sigma_i^x - J \sum_i \sigma_i^z \tau_{i,i+1}^z \sigma_{i+1}^z - J_v \sum_i \tau_{i-1,i}^x \sigma_i^x \tau_{i,i+1}^x. \tag{35}$$

When $J = 0$, in the ground state, all the $\sigma$ spins are polarized in the $X$ direction and the gauge fields couple effectively through $\tau_{i-1,i}^x \tau_{i,i+1}^x$. With respect to the effective global symmetry of $B = \prod_i \tau_{i,i+1}^z$, the gauge field ground state spontaneously breaks the symmetry.

On the other hand, if $B_x = 0$, the Hamiltonian becomes a 1D cluster state [54] model with unique ground state which is symmetric under the global $B = \prod_i \tau_{i,i+1}^z$ symmetry.

Now let us discuss an SPT example. Consider the 1D cluster state model

$$H = -\sum_i h_{2i-1}^o - \sum_i h_{2i}^e = -\sum_i \sigma_{2i-2}^z \sigma_{2i-1}^x \sigma_{2i}^z - \sum_i \sigma_{2i-1}^z \sigma_{2i}^x \sigma_{2i+1}^z. \tag{36}$$

This model has a global $Z_2 \times Z_2$ symmetry generated by

$$g_1 = \prod_i \sigma_{2i}^x, \qquad\qquad g_2 = \prod_i \sigma_{2i-1}^x \tag{37}$$

and the model has symmetry protected topological order under this symmetry [55].

To gauge the $Z_2 \times Z_2$ symmetry, we put gauge fields $\tau$ between neighboring gauge charges. That is, we place one gauge DOF per site. The ones on the even sites are gauge fields of $g_2$. The ones on the odd sites are gauge fields of $g_1$. The Gauss law terms are

$$c_{2i} = \tau_{2i-1}^x \sigma_{2i}^x \tau_{2i+1}^x, \qquad\qquad c_{2i+1} = \tau_{2i}^x \sigma_{2i+1}^x \tau_{2i+2}^x. \tag{38}$$

The flux terms, which are pure gauge terms that satisfy the Gauss law, are

$$f_1 = \prod_i \tau_{2i-1}^z, \qquad\qquad f_2 = \prod_i \tau_{2i}^z. \tag{39}$$

They become the global $Z_2 \times Z_2$ symmetry of the gauged model.

To make the original Hamiltonian terms gauge invariant, we modify them to be

$$h^o_{2i-1} = \sigma^z_{2i-2}\sigma^x_{2i-1}\tau^z_{2i-1}\sigma^z_{2i}, \qquad\qquad h^e_{2i} = \sigma^z_{2i-1}\sigma^x_{2i}\tau^z_{2i}\sigma^z_{2i+1}. \qquad (40)$$

Now the total Hamiltonian is

$$H_g = -\sum_i \left( \tau^x_{2i-1}\sigma^x_{2i}\tau^x_{2i+1} + \tau^x_{2i}\sigma^x_{2i+1}\tau^x_{2i+2} + \sigma^z_{2i-2}\sigma^x_{2i-1}\tau^z_{2i-1}\sigma^z_{2i} + \sigma^z_{2i-1}\sigma^x_{2i}\tau^z_{2i}\sigma^z_{2i+1} \right). \qquad (41)$$

All the terms commute, are independent, and are symmetric under the global symmetry. Therefore, on a closed ring, the ground state is unique. On an open chain, the terms

$$\sigma^x_1 \tau^x_2, \qquad \tau^x_{2N-1}\sigma^x_{2N} \qquad (42)$$

no longer commute with the symmetry and need to be removed, leaving a two fold degeneracy at the edge as the symmetry protected edge state.

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
