# Peer review of "Foliated fracton order from gauging subsystem symmetries"

_SciPost Physics, doi:SciPost Phys. 6, 041 (2019)_

## Round 1 · Referee Report · Anonymous (Referee 1) · 2018-11-27

Strengths

1) Very clear exposition of the gauging procedure

2) Connections between subsystem symmetry charges and behavior of gauged system's excitations is useful and clear

3) Contains some novel and potentially useful examples

Weaknesses

1) The genericness of the correspondence between properties of the ungauged and gauged systems is often either unclear or unsupported by evidence.

2) The purpose of each example in the paper, of which there are many, is not always clear.

Report

This work explains the general procedure for gauging subsystem symmetries and then explains how features of ungauged models, such as spontaneous symmetry breaking or objects carrying specific symmetry charges, lead to specific features of their gauged counterparts.

The gauging procedure itself is essentially a review of the procedure from Vijay et. al. (Ref. 3 in the present paper), although it makes a few generalities more explicit and adds some new examples. That said, the exposition of the gauging procedure is extremely clear.

Section 4 is the primary novel portion of this work. The connection between charge under various subsystem symmetries in the ungauged system and the mobility of excitations in the gauged system is made very clear and explicit. I see this as a useful way of thinking in a model-independent way about how objects of varying mobility emerge from the gauging procedure.

Along these lines, I would encourage the authors to more clearly emphasize the genericness of behaviors when appropriate, as it is not always clear whether or not some of the features of the gauge correspondence are model-dependent. For example, in sections 4.7-4.8, while the self-duality of these models is interesting, is the intent to claim that gauging *any* SSPT with linear symmetries implements a self-duality? Or is it a claim about particular exactly solvable models? If a claim is being made that such behaviors are generic, that should be clarified and justified, and if not, this point should be made clearer.

Presentation-wise, there are ten examples in section 4. That is a lot, given that many of them appear to be illustrating essentially the same point. It would be very helpful to more clearly explain what each example is intended to convey. Alternatively, if section 4 is intended to exhaust some set of possibilities, it should be made clear what set is being exhausted (specific lattices? specific symmetry actions?).

Requested changes

1) Add discussion about which features of which examples are generic and which are model-specific (see report)

2) Better signposting in section 4 (see report)

3) I think it is worth explicitly pointing out that from this point of view, the gauge structure, that is, the gauge degrees of freedom and the operator which determines the gauge symmetry, is completely independent of the original Hamiltonian and depends only on the action of the symmetry operators on the ungauged Hilbert space. (This would help distinguish the point of view of this paper from Ref. 3, which takes a strongly Hamiltonian-centric view.)

4) In Fig. 5a, I believe that the label $\sigma_c$ should read $\sigma_a$ to be consistent with the definitions above Eq. (8).

5) The accuracy of the discussion just prior to section 4.1 about composing fracton charges depends on the charges involved when the symmetries are not $\mathbb{Z}_2$. A qualifier about the relevant symmetry charges should be added.

6) Table 1 would be more comprehensive if it indicated that the correspondence between symmetry-broken states and trivial states also holds for 3D planar symmetries.

7) (optional) If the authors would prefer to gauge $U(1)$ subsystem symmetries rather than global symmetries in section 3.6, they could use the (0,1) a.k.a. hollow rank-2 scalar charge theory (in the language of Refs. 50 and 51 respectively) instead of the conventional scalar charge theory.

  • validity: high
  • significance: good
  • originality: ok
  • clarity: high
  • formatting: excellent
  • grammar: perfect

Author:  Xie Chen  on 2019-01-13  [id 402]

(in reply to Report 1 on 2018-11-27)

We are very grateful to the insightful comments of the referee. We have made changes accordingly as listed below.

1) Add discussion about which features of which examples are generic and which are model-specific (see report)

2) Better signposting in section 4 (see report)

We thank the referee for the helpful suggestions. To address these issues, we have made the following changes:

a. we have separated the examples in section 4 into two subsections, one on planar symmetries and one on linear symmetries and added the following sentence to explain the logic behind these models:

We are going to study models of increasing complexity – paramagnets with subsystem planar symmetries in one direction, two directions, three directions and four directions respectively – as well as models where the symmetries are spontaneously broken.

b. we have added the following discussion to section 4.1 on planar symmetries:

We expect the following features to be generically true in the gauging correspondence: 1. when the planar symmetries are spontaneously broken, the gauged model does not have nontrivial order; 2. when the planar symmetries are not spontaneously broken, the gauged model has foliated fracton order 3. symmetry charges transforming under planar symmetries in one direction, two directions, and three or more directions turn into planon excitations, lineon excitations, and fracton excitations respectively upon gauging. The first feature is analogous to the Higgs mechanism in usual gauge theories. For the second one, we gave an intuitive understanding in the introduction section. Let us briefly discuss the third one before moving onto examples.

c. we have added the following discussion to section 4.2 on linear symmetries:

We find that the gauging correspondence works in a very similar way to that of linear symmetries in 1D where it becomes a global symmetry. It is well known (and we review it in Appendix \ref{sec:gauge1D}) that upon gauging the linear (global) symmetry in 1D, the gauged model also has an emergent global linear symmetry at low energy which comes from the zero flux constraint around the 1D ring. The gauging procedure leads to a duality between trivial symmetric paramagnets and symmetry breaking phases and a (self)-duality among nontrivial symmetry protected topological phases. From the examples discussed in this section, we find a similar correspondence in 2D and 3D with subsystem linear symmetries: 1. the model after gauging has linear subsystem symmetries at low energy which comes from the zero flux constraint around nontrivial loops; 2. symmetry breaking phases are mapped to trivial paramagnets; 3. trivial paramagnets are mapped to symmetry breaking phases; 4. non-trivial subsystem symmetry protected topological phases are mapped to non-trivial subsystem symmetry protected topological phases. We expect these features to apply generically to all models with linear subsystem symmetries.

3) I think it is worth explicitly pointing out that from this point of view, the gauge structure, that is, the gauge degrees of freedom and the operator which determines the gauge symmetry, is completely independent of the original Hamiltonian and depends only on the action of the symmetry operators on the ungauged Hilbert space. (This would help distinguish the point of view of this paper from Ref. 3, which takes a strongly Hamiltonian-centric view.)

This is a great suggestion. We have added the following sentence to section 3.1:

Note that a large part this procedure, such as determining the gauge degrees of freedom and the gauge symmetry, is completely independent of the original Hamiltonian and depends only on the action of the symmetry operators on the ungauged Hilbert space. The only step that depends on the original Hamiltonian is step 4 where the original Hamiltonian is made gauge symmetric.

4) In Fig. 5a, I believe that the label σc should read σa to be consistent with the definitions above Eq. (8).

We would like to thank the referee for his / her careful reading. We have changed the text around Eq. (8) to be consistent with Fig. 5a.

5) The accuracy of the discussion just prior to section 4.1 about composing fracton charges depends on the charges involved when the symmetries are not Z2. A qualifier about the relevant symmetry charges should be added.

Thanks for pointing this out. We have made the changes accordingly.

6) Table 1 would be more comprehensive if it indicated that the correspondence between symmetry-broken states and trivial states also holds for 3D planar symmetries.

Thanks. We have added that to the table.

7) (optional) If the authors would prefer to gauge U(1) subsystem symmetries rather than global symmetries in section 3.6, they could use the (0,1) a.k.a. hollow rank-2 scalar charge theory (in the language of Refs. 50 and 51 respectively) instead of the conventional scalar charge theory.

Thanks. We decided to keep section 3.6 about the conventional scalar charge theory to show that the gauge procedure works generally. To avoid conflict with the title of section 3, we moved this section to the appendix.

---

## Round 1 · Referee Report · Anonymous (Referee 2) · 2018-12-1

Strengths

1 - This paper constructs a clear general procedure for gauging subsystem symmetries, which had previously only been done on a case-by-case basis.
2 - The paper provides an excellent review of standard gauging procedures.
3 - The paper draws close connections to other topics in fracton physics, such as foliation, SSPTs, and tensor gauge theories.

Weaknesses

1 - The discussion on tensor gauge theories seems to come out of nowhere (see below).

Report

In this work, Shirley, Slagle, and Chen construct a very general procedure for gauging subsystem symmetries, which is a useful concept for obtaining fracton models. Their procedure works for a wide range of geometries, many of which they carry through explicitly, while previous work had done things on a case-by-case basis. The analysis covers both planar and linear subsystem symmetries. Furthermore the authors show that there is a direct correspondence between these subsystem symmetries and the properties of excitations of the resulting gauged phases. For example, the mobility of a particle is dictated by the way in which that charge transforms under various planar symmetries. While these results are not unexpected, it is good to see how they arise in a systematic way. I therefore recommend publication on SciPost.

Requested changes

1 - Section 3.6 on tensor gauge theories seems to appear rather abruptly. It is satisfying that the procedure for gauging subsystem symmetries seems to work for these theories, but why does it work in this case? Is there some connection between tensor gauge theories and subsystem symmetries? Either way, a bit more motivation in this section would be useful.

  • validity: high
  • significance: good
  • originality: good
  • clarity: high
  • formatting: excellent
  • grammar: excellent

Author:  Xie Chen  on 2019-01-13  [id 403]

(in reply to Report 2 on 2018-12-01)

We would like to thanks the referee for the helpful comment. Section 3.6 does indeed seem out of place because it studies a system with a global symmetry rather than subsystem symmetries. The gauging procedure discussed in section 3.1 applies to global symmetries and subsystem symmetries alike, so it should work for this example. We added this section mostly to show how the gauging procedure works for continuous symmetry (all other examples in the paper involve discrete symmetries) and show that this procedure can produce the tensor gauge theory which has been extensively studied in the fracton literature. To resolve the disconnectedness of topic, we have moved this section to the appendix.

---

## Round 2 · Referee Report · Anonymous · 2019-3-21

Report

The authors have now reorganized the text in several ways to make the presentation more logical, and they have addressed my earlier comments on tensor gauge theories. The paper is well-written and of very good scientific quality. As such, I can recommend its acceptance for publication by SciPost.

---

## Round 2 · Author Response

We would like to thank the editor and referees for reviewing our manuscript. We have replied to the referee reports using SciPost comments and have improved our manuscript with their suggestions.

---

## Round 2 · List of Changes

1) As suggested, we added some discussion to the end of Section 3.1.
2) We improved the signposting in Section 4 by reorganized the examples into two subsections with additional introductory text at the beginning of each subsection.
3) A row was added to Table 1.
4) Appendix B was moved from the main text to the Appendix.

---

## Editorial Decision

published